# Visual impairment among diabetes patients in Ethiopia: A systematic review and meta-analysis

Tigabu Munye Aytenew[1]*, Demewoz Kefale[2], Binyam Minuye Birhane[3,4], Solomon Demis Kebede[4], Worku Necho Asferie[4], Habtamu Shimels Hailemeskel[4], Amare Kassaw[2], Sintayehu Asnakew[5], Yohannes Tesfahun Kassie[6], Gebrehiwot Berie Mekonnen[2], Melese Kebede[6], Yeshiambaw Eshetie[1], Netsanet Ejigu[7], Shegaw Zeleke[1], Muluken Chanie Agimas[8], Amare Simegn[9]

1 Department of Nursing, College of Health Sciences, Debre Tabor University, Debre Tabor, Ethiopia, 2 Department of Pediatrics and Child Health Nursing, College of Health Sciences, Debre Tabor University, Debre Tabor, Ethiopia, 3 School of Public Health, University of Technology Sydney, Sydney, NSW, Australia, 4 Department of Maternity and Neonatal Nursing, College of Health Sciences, Debre Tabor University, Debre Tabor, Ethiopia, 5 Department of Psychiatry, College of Health Sciences, Debre Tabor University, Debre Tabor, Ethiopia, 6 Department of Emergency and Critical Care Nursing, College of Health Sciences, Debre Tabor University, Debre Tabor, Ethiopia, 7 Department of Midwifery, Dembya Primary Hospital, Koladiba, Gondar, Ethiopia, 8 Department of Epidemiology and Biostatistics, Institute of Public Health, College of Medicine and Health Science, University of Gondar, Gondar, Ethiopia, 9 Department of Reproductive Health, College of Health Sciences, Debre Tabor University, Debre Tabor, Ethiopia

* tigabumunye21@gmail.com

**Data Availability Statement:** All relevant data are within the paper and its Supporting Information files.

## Abstract

### Introduction

The increased prevalence of visual impairment among diabetes patients has become a major global public health problem. Although numerous primary studies have been conducted to determine the prevalence of visual impairment and its associated factors among diabetes patients in Ethiopia, these studies presented inconsistent findings. Therefore, this review aimed to determine the pooled prevalence of visual impairment and identify associated factors among diabetes patients.

### Methods

An extensive search of literature was done on PubMed, Google Scholar, and Web of Sciences databases. A manual search of the reference lists of included studies was performed. A weighted inverse-variance random-effects model was used to calculate the pooled prevalence of visual impairment.

### Results

A total of 34 eligible primary studies with a sample size of 11,884 participants were included in the final meta-analysis. The pooled prevalence of visual impairment was 21.73% (95% CI: 18.15, 25.30; $I^2$ = 96.47%; P<0.001). Diabetes mellitus with a duration of diagnosis ≥10 years [AOR = 3.18, 95% CI: 1.85, 5.49], presence of co-morbid hypertension [AOR = 3.26,

**Funding:** The author(s) received no specific funding for this work.

**Competing interests:** The authors have declared that no competing interests exist.

**Abbreviations:** DM, Diabetes mellitus; DR, Diabetic retinopathy; ICAM, Inter-cellular adhesion molecule; IDF, International Diabetic Federation; LMICs, Low and middle-income countries; NCDs, Non-communicable diseases; PRISMA, Preferred Reporting Items for Systematic Reviews and Meta-Analyses; VI, Visual impairment.

95% CI: 1.93, 5.50], poor glycemic control [AOR = 4.30, 95% CI: 3.04, 6.06], age ≥56 years [AOR = 4.13, 95% CI: 2.27, 7.52], family history of diabetes mellitus [AOR = 4.18 (95% CI: 2.61, 6.69], obesity [AOR = 4.77, 95% CI: 3.00, 7.59], poor physical activity [AOR = 2.46, 95% CI: 1.75, 3.46], presence of visual symptoms [AOR = 4.28, 95% CI: 2.73, 6.69] and no history of eye exam [AOR = 2.30, 95% CI: 1.47, 3.57] were significantly associated with visual impairment.

## Conclusions

The pooled prevalence of visual impairment was high in Ethiopia. Diabetes mellitus with a duration of diagnosis ≥10 years, presence of co-morbid hypertension, poor glycemic control, age ≥56 years, and family history of diabetes mellitus, obesity, poor physical activity, presence of visual symptoms, and no history of eye exam were independent predictors. Therefore, diabetic patients with these identified risks should be screened, and managed early to reduce the occurrence of visual impairment related to diabetes. Moreover, public health policy with educational programs and regular promotion of sight screening for all diabetes patients are needed.

## Introduction

Diabetes mellitus (DM) is a major global public health problem [1]. It was one of the four priority non-communicable diseases (NCDs) targeted for prevention and control in 2011 [2]. According to the International Diabetic Federation's (IDF) 2019 report, it was estimated that around 500 million people are living with diabetes worldwide [3], and predicted to be 693 million by 2045 [4]. The majority occurred in low and middle-income countries (LMICs) [5], and 2.6 million diabetes cases were also reported in Ethiopia by 2017 [4].

DM is associated with chronic complications like diabetic neuropathy, nephropathy, retinopathy, cardiovascular diseases, blindness, kidney failure, and nerve damage [6, 7]. It causes visual impairment (VI) through early-onset cataracts and diabetic retinopathy (DR), a progressive disease of the retinal microvasculature [8]. Globally, around 2.2 billion people have a near or distant visual impairment, of whom 3.9 million are visually impaired due to diabetic retinopathy [9]. In Africa, the prevalence of visual impairment among diabetes patients ranges from 17.1% to 78.25% [10–12]. The increased prevalence of diabetes-related visual impairment has become a major global public health problem requiring substantial attention [13–15]. It is more common among people with diabetes than in people without diabetes [16, 17]. Visual impairment among diabetes patients can be associated with older age, poor glycemic control, poor physical exercise, long durations of diabetes, and type of treatment [18–20].

Visual impairment can increase the unemployment rate and medical expenses, and reduce the performance of daily living activities, productivity, and social participation, leading an individual with diabetes to have a reduced quality of life [9, 21]. Therefore, controlling blood glucose levels, regular physical activity, having regular eye exams, and undergoing early laser photocoagulation have been used to reduce the burden of visual impairment among diabetes patients [22–24].

Although numerous primary studies have been conducted to determine the prevalence of visual impairment and its associated factors among diabetes patients in Ethiopia, these studies presented inconsistent findings, ranging from 7% [25] to 42% [26]. Therefore, this review

aimed to determine the pooled prevalence of visual impairment and identify associated factors.

# Methods

## Reporting and registration protocol

The Preferred Reporting Items for Systematic Reviews and Meta-Analyses (PRISMA) statement guideline [27] was used to report the results of this systematic review and meta-analysis (S1 Table). The review protocol was registered with Prospero database: (PROSPERO, 2023: CRD42023438607).

## Databases and search strategy

We have conducted a thorough search of databases on PubMed, Google Scholar, and Web of Sciences for all relevant studies conducted in Ethiopia using the following search terms and phrases: (″Prevalence″ OR ″Incidence″ OR ″Magnitude″ OR ″Burden″) AND (″Visual impairment″) OR ″Diabetic retinopathy″ OR ″Retinopathy″ OR ″Cataract″ AND (″Associated factors″ OR ″Determinant factors″ OR ″Risk factors″ OR ″Determinants″) AND ″Ethiopia″. Besides, a manual search of the reference lists of included studies was performed. The searched primary studies were published in the English language between 2011 and 2023 in Ethiopia.

## Eligibility criteria

All observational studies which were conducted among diabetes patients in Ethiopia, and reported the prevalence of visual impairment, associated factors, and written in English were included in the review. However, citations without abstracts, full texts, anonymous reports, editorials, systematic reviews and meta-analyses, and qualitative studies were excluded from the review.

## Study selection

All the retrieved studies were exported to the EndNote version 7 reference manager to remove duplicate studies. Initially, two independent reviewers (TMA and DK) screened the titles and abstracts, followed by the full-text reviews to determine the eligibility of each study. The disagreement between the two reviews was solved through dialogue.

## Data extraction

Two independent reviewers (TMA and AS) extracted the data using structured Microsoft Excel. When variations were observed in the extracted data, the phase was repeated. If discrepancies between the extracted data continued, the third reviewer (SDK) was involved. The name of the first author, year of publication, study area, study design, sample size, response rate, and effect size of the eligible studies were collected.

## Primary outcome measure

The primary outcome of interest was the pooled prevalence of visual impairment among diabetes patients in Ethiopia.

## Data analysis

The extracted data were exported to STATA version 17 for statistical analysis. A weighted inverse-variance random-effects model [28] was used to calculate the pooled prevalence of

visual impairment and determine the impact of its associated factors. The presence of publication bias was checked by observing the symmetry of the funnel plot and Egger's test with a p-value of <0.05 was employed to determine significant publication bias [29]. The percentage of total variation across studies due to heterogeneity was assessed using $I^2$ statistical test [30]. The $I^2$ values of 0, 25, 50, and 75% represented no, low, moderate, and high heterogeneity respectively [30].

A p-value of $I^2$ statistic <0.05 was used to declare a significant heterogeneity [31, 32]. To identify the influence of a single study on the overall meta-analysis, sensitivity analysis was performed. A forest plot was used to estimate the effect of independent factors on the outcome variable and a measure of association at 95% CI was reported. The adjusted odds ratio (AOR) was the most commonly reported measure of association in the eligible primary studies, and a random-effects model was used to estimate the pooled OR effect.

## Results

### Search results

A total of 2476 studies were retrieved from PubMed (n = 1294), Google Scholar (n = 1127), Web of Science (n = 39) databases, manual search (n = 7) and the University's research repository online library (n = 9). Upon removing the duplicated studies (n = 129) and irrelevant studies based on their titles and abstracts (n = 1852), a total of 495 studies were selected for full-text review. During full-text review, 382 studies with no accessible full texts were removed. Of the remaining 113 studies, 79 studies were excluded (full texts were not written in English, different settings, and the outcomes were not well defined). Finally, 34 studies were extracted to determine the pooled prevalence of visual impairment and its associated factors among diabetes patients in Ethiopia. We traced the PRISMA flow chart [33] to show the selection process from initially identified records to finally included primary studies (Fig 1).

### Characteristics of the included studies

The twenty-six studies [20, 25, 26, 34–56], six studies [57–62] and two studies [63, 64] were conducted using cross-sectional, retrospective cohort and case-control study designs respectively. Regarding geographical region, seventeen studies [20, 25, 26, 35, 36, 38, 40, 43, 45, 49–51, 55, 56, 60–62] were conducted in Amhara, eight studies [39, 41, 44, 48, 52, 58, 59, 63] were conducted in Oromia, five studies [42, 46, 53, 57, 64] were conducted in Addis Ababa, three studies [37, 47, 54] were conducted in Southern nations, nationalities and peoples, and one study [34] was conducted in Sidama region.

The total sample size of the included studies was 11,884, where the smallest sample size was 81 [54] in Southern nations, nationalities and Peoples, and the largest sample size 739 [51] in Amhara region. The prevalence of visual impairment among diabetes patients was obtained from thirty-two included primary studies [20, 25, 26, 34–62], while the data regarding the associated factors of visual impairment were obtained from the twenty-two studies [20, 34, 35, 37–40, 43–47, 49, 50, 55, 56, 58–60, 62–64], with a response rate ranges from 89.33 to 100% (Table 1).

### Operational definition of variables

Visual impairment is the loss of the functionality of the visual systems, characterized by decreased visual acuity, visual field loss, visual distortion, or perception problems [34, 36, 65, 66].

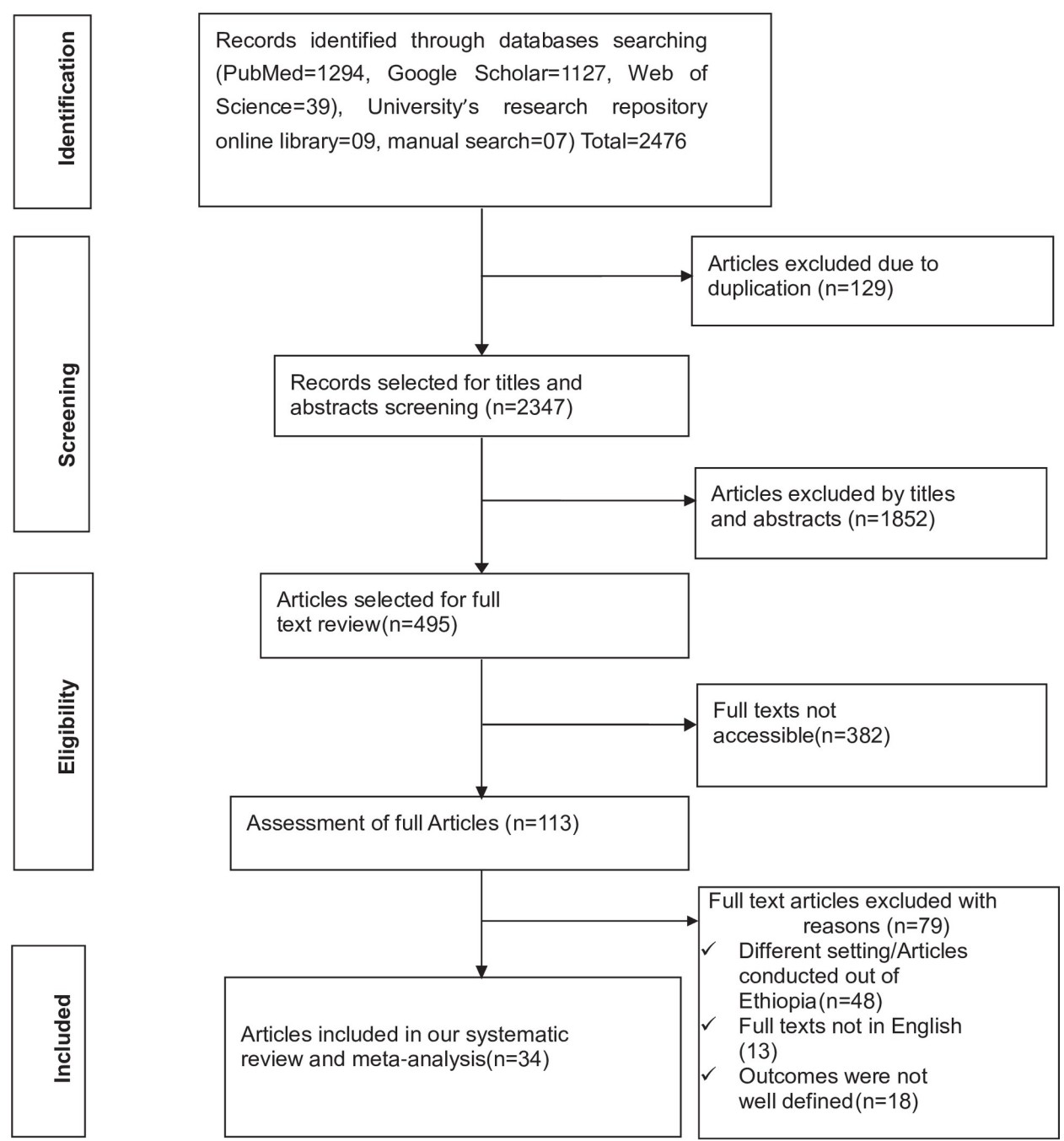

**Fig 1. PRISMA flow chart showing the studies selection process, 2023.**

## Quality appraisal of the included studies

Two independent reviewers (TMA and DK) appraised the quality of the included studies and scored for the validity of the results. The quality of each study was evaluated using the Joanna Briggs Institute (JBI) quality appraisal criteria [67]. Twenty-six studies [20, 25, 26, 34–56], six studies [57–62] and two studies [63, 64] were appraised using JBI checklist for cross-sectional, cohort and case-control studies, respectively.

**Table 1. General characteristics of the included primary studies, 2023.**

| ID | Author (Year) | Study area | Study design | Sample size | Prevalence (95% CI) | Quality |
|---|---|---|---|---|---|---|
| | Alemayehu HB [2022] [34] | Sidama | CS | 398 | 28.60(24.16, 33.04) | Low risk |
| | Alemu Mersha G [2020] [35] | Amhara | CS | 306 | 15.30(11.27, 19.33) | Low risk |
| | Alemu S [2015] [25] | Amhara | CS | 544 | 7.0(4.86, 9.14) | Low risk |
| | Asemu MT [2021] [36] | Amhara | CS | 401 | 12.46(9.23, 15.69) | Low risk |
| | Azeze TK [2018] [57] | A. A | Cohort | 377 | 18.57(14.65, 22.49) | Low risk |
| | Chisha Y [2017] [37] | SNNP | CS | 400 | 13.0(9.70, 16.30) | Low risk |
| | Debele GR [2021] [58] | Oromia | Cohort | 402 | 20.15(16.23, 24.07) | Low risk |
| | Demilew KZ [2022] [38] | Amhara | CS | 388 | 29.38(24.85, 33.91) | Low risk |
| | Ejeta A [2021] [39] | Oromia | CS | 319 | 13.0(9.31, 16.69) | Low risk |
| | Ejigu T [2021] [40] | Amhara | CS | 225 | 10.70(6.66, 14.74) | Low risk |
| | Garoma D [2020] [63] | Oromia | Case-control | 311 | Not applicable | Low risk |
| | Gelcho GN [2022] [59] | Oromia | Cohort | 373 | 41.30(36.30, 46.30) | Low risk |
| | Gizaw M [2015] [42] | A. A | CS | 523 | 11.0(8.32, 13.68) | Low risk |
| | Gudina EK [2011] [41] | Oromia | CS | 329 | 23.10(18.55, 27.65) | Low risk |
| | Kabtu E [2022] [26] | Amhara | CS | 165 | 42.0(34.47, 49.53) | Low risk |
| | Kebede SA [2022] [60] | Amhara | Cohort | 489 | 17.17(13.83, 20.51) | Low risk |
| | Lebeta R [2017] [43] | Amhara | CS | 344 | 25.50(20.89, 30.11) | Low risk |
| | Sahiledengle B [2022] [44] | Oromia | CS | 256 | 19.90(15.01, 24.79) | Low risk |
| | Seid K [2021] [64] | A. A | Case-control | 282 | Not applicable | Low risk |
| | Seid MA [2021] [45] | Amhara | CS | 335 | 24.80(20.18, 29.42) | Low risk |
| | Seid MA [2022] [20] | Amhara | CS | 322 | 37.58(32.29, 42.87) | Low risk |
| | Shibru T [2019] [46] | A. A | CS | 191 | 51.30(44.21, 58.39) | Low risk |
| | Takele MB [2022] [61] | Amhara | Cohort | 494 | 4.80(2.92, 6.69) | Low risk |
| | Tesfaye DJ [2015] [47] | SNNP | CS | 266 | 11.70(7.84, 15.56) | Low risk |
| | Tilahun AN [2017] [48] | Oromia | CS | 236 | 20.30(15.17, 25.43) | Low risk |
| | Tilahun M [2020] [49] | Amhara | CS | 302 | 18.90(14.48, 23.32) | Low risk |
| | Tilahun M [2021] [50] | Amhara | CS | 426 | 26.30(22.12, 30.48) | Low risk |
| | Tsegaw A [2021] [51] | Amhara | CS | 739 | 10.60(8.38, 12.82) | Low risk |
| | Wolde HF [2018] [62] | Amhara | Cohort | 341 | 18.40(14.29, 22.51) | Low risk |
| | Worku D [2010] [52] | Oromia | CS | 305 | 33.80(28.49, 39.11) | Low risk |
| | Woyessa DN [2020] [53] | A. A | CS | 111 | 21.60(13.95, 29.26) | Low risk |
| | YimamAhmed M [2020] [54] | SNNP | CS | 100 | 8.50(3.03, 13.97) | Low risk |
| | Zegeye AF [2023] [55] | Amhara | CS | 496 | 36.30(32.07, 40.53) | Low risk |
| | Zewdu K [2017] [56] | Amhara | CS | 388 | 29.38(24.85, 33.91) | Low risk |

Abbreviations: A.A, Addis Ababa; CS, cross-sectional; SNNP, Southern nations, nationalities and peoples.

Thus, among the twenty-six cross-sectional studies, twenty-one studies scored seven of the eight questions, 87.5% (low risk), three studies scored six of the eight questions, 75% (low risk), and the remaining two studies also scored five of the eight questions, 62.5% (low risk). But the two cross-sectional studies [68, 69] were appraised, and each scored three of the eight questions, 37.5% (high risk). As a result, these two studies have been excluded from the study due to their low quality. Likewise, among the six cohort studies, four studies scored eight of the ten questions, 80% (low risk), and two studies also scored seven of the ten questions, 70% (low risk). Moreover, the two case-control studies were appraised, and each study scored eight of the ten questions (S2 Table in S1 File).

Studies were of low risk when they scored 50% or higher on the quality assessment indicators. After conducting a thorough quality appraisal, we determined that the primary studies

included in the analysis displayed a high level of reliability in their methodological quality scores. The cross-sectional studies scored between 5 and 7 out of a total of 8 points, while the cohort and case-control studies scored between 7 and 8 out of a total of 10 points. Hence, all the included primary studies [20, 25, 26, 34–64] had high quality.

**Risk of bias assessment.** The adopted assessment tool [70] was used to assess the risk of bias. The tool consists of ten items that assess four areas of bias: internal validity and external validity. Items 1–4 evaluate selection bias, non-response bias and external validity. Items 5–10 assess measure bias, analysis-related bias, and internal validity. Accordingly, of the total of the thirty-four included studies, twenty-nine studies scored eight of the ten questions and five studies also scored seven of the ten questions. Studies were classified as ″low risk″ if eight and above of the ten questions received ″Yes″, as ″moderate risk″ if six to seven of the ten questions received ″Yes″ and as ″high risk″ if five or lower of the ten questions received ″Yes″. Therefore, all the included primary studies [20, 25, 26, 34–64] had a low risk of bias (high quality) (S3 Table in S1 File).

## Meta-analysis

### Pooled prevalence of visual impairment

A total of 34 eligible primary studies [20, 25, 26, 34–64] were included in the final meta-analysis, and the pooled prevalence of visual impairment among diabetes patients in Ethiopia was 21.73% (95% CI:18.15, 25.30; $I^2$ = 96.47%; P<0.001) (Fig 2).

### Publication bias

The asymmetric distribution of the included primary studies on the funnel plot suggests the presence of publication bias (Fig 3A), and the p-value of Egger's regression test (P<0.001) also indicated the presence of publication bias. Hence, trim and fill analyses were done to manage the publication bias (Fig 3B).

### Investigation of heterogeneity

The percentage of $I^2$ statistics of the forest plot indicates a marked heterogeneity among the included primary studies ($I^2$ = 96.47%, P<0.001) (Fig 2). Hence, sensitivity and subgroup analyses were performed to minimize the heterogeneity.

### Sensitivity analysis

To determine the influence of a particular primary study on the overall meta-analysis, sensitivity analysis was conducted. The forest plot showed that the estimate from a single primary study is closer to the combined estimate, which implied the absence of a single study effect on the overall pooled estimate. Thus, it has been demonstrated that a single study has no significant impact on the overall outcome of the meta-analysis (Fig 4).

### Subgroup analysis

The subgroup analysis was performed based on the study area and study period. Thus, the highest pooled prevalence of visual impairment was found among studies conducted in Addis Ababa [25.35, 95% CI: 11.18, 39.52, $I^2$ = 97.30%, P<0.001], followed by studies conducted in Oromia region [24.42, 95% CI: 17.38, 31.47, $I^2$ = 94.02%, P<0.001] (Fig 5). Similarly, the higher pooled prevalence of visual impairment was among studies conducted in the year 2021 and later [23.25, 95% CI: 17.58, 28.91; $I^2$ = 96.39%, P<0.001], followed by studies conducted before the year 2021 [20.19, 95% CI: 15.66, 24.73, $I^2$ = 95.05%, P<0.001] (Fig 6). Based on the

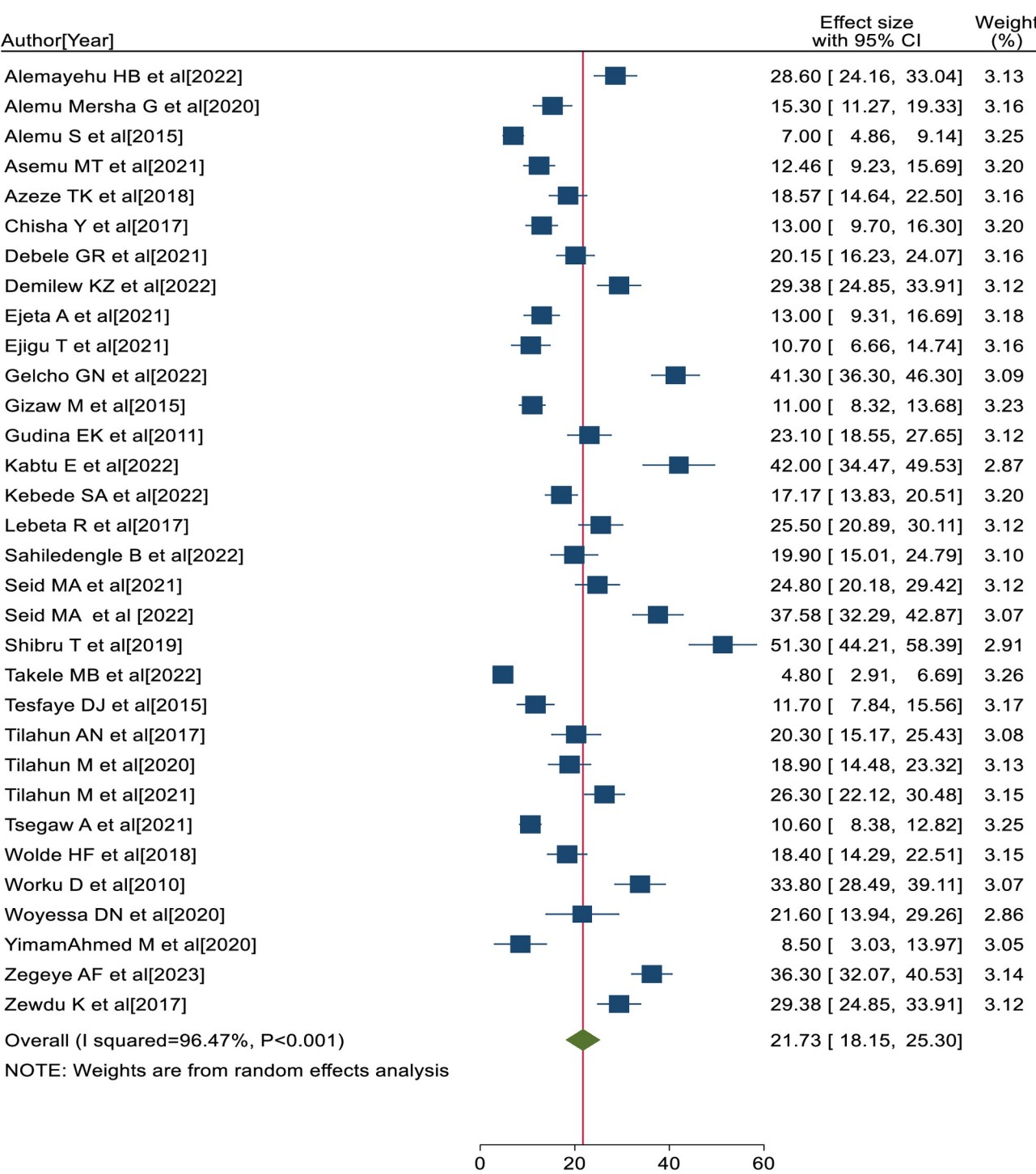

| Author[Year] | Effect size with 95% CI | Weight (%) |
|---|---|---|
| Alemayehu HB et al[2022] | 28.60 [ 24.16, 33.04] | 3.13 |
| Alemu Mersha G et al[2020] | 15.30 [ 11.27, 19.33] | 3.16 |
| Alemu S et al[2015] | 7.00 [ 4.86, 9.14] | 3.25 |
| Asemu MT et al[2021] | 12.46 [ 9.23, 15.69] | 3.20 |
| Azeze TK et al[2018] | 18.57 [ 14.64, 22.50] | 3.16 |
| Chisha Y et al[2017] | 13.00 [ 9.70, 16.30] | 3.20 |
| Debele GR et al[2021] | 20.15 [ 16.23, 24.07] | 3.16 |
| Demilew KZ et al[2022] | 29.38 [ 24.85, 33.91] | 3.12 |
| Ejeta A et al[2021] | 13.00 [ 9.31, 16.69] | 3.18 |
| Ejigu T et al[2021] | 10.70 [ 6.66, 14.74] | 3.16 |
| Gelcho GN et al[2022] | 41.30 [ 36.30, 46.30] | 3.09 |
| Gizaw M et al[2015] | 11.00 [ 8.32, 13.68] | 3.23 |
| Gudina EK et al[2011] | 23.10 [ 18.55, 27.65] | 3.12 |
| Kabtu E et al[2022] | 42.00 [ 34.47, 49.53] | 2.87 |
| Kebede SA et al[2022] | 17.17 [ 13.83, 20.51] | 3.20 |
| Lebeta R et al[2017] | 25.50 [ 20.89, 30.11] | 3.12 |
| Sahiledengle B et al[2022] | 19.90 [ 15.01, 24.79] | 3.10 |
| Seid MA et al[2021] | 24.80 [ 20.18, 29.42] | 3.12 |
| Seid MA et al [2022] | 37.58 [ 32.29, 42.87] | 3.07 |
| Shibru T et al[2019] | 51.30 [ 44.21, 58.39] | 2.91 |
| Takele MB et al[2022] | 4.80 [ 2.91, 6.69] | 3.26 |
| Tesfaye DJ et al[2015] | 11.70 [ 7.84, 15.56] | 3.17 |
| Tilahun AN et al[2017] | 20.30 [ 15.17, 25.43] | 3.08 |
| Tilahun M et al[2020] | 18.90 [ 14.48, 23.32] | 3.13 |
| Tilahun M et al[2021] | 26.30 [ 22.12, 30.48] | 3.15 |
| Tsegaw A et al[2021] | 10.60 [ 8.38, 12.82] | 3.25 |
| Wolde HF et al[2018] | 18.40 [ 14.29, 22.51] | 3.15 |
| Worku D et al[2010] | 33.80 [ 28.49, 39.11] | 3.07 |
| Woyessa DN et al[2020] | 21.60 [ 13.94, 29.26] | 2.86 |
| YimamAhmed M et al[2020] | 8.50 [ 3.03, 13.97] | 3.05 |
| Zegeye AF et al[2023] | 36.30 [ 32.07, 40.53] | 3.14 |
| Zewdu K et al[2017] | 29.38 [ 24.85, 33.91] | 3.12 |
| Overall (I squared=96.47%, P<0.001) | 21.73 [ 18.15, 25.30] | |

NOTE: Weights are from random effects analysis

Random-effects DerSimonian–Laird model

**Fig 2. Forest plot showing the pooled prevalence of visual impairment with 95% CIs in Ethiopia, 2023.**

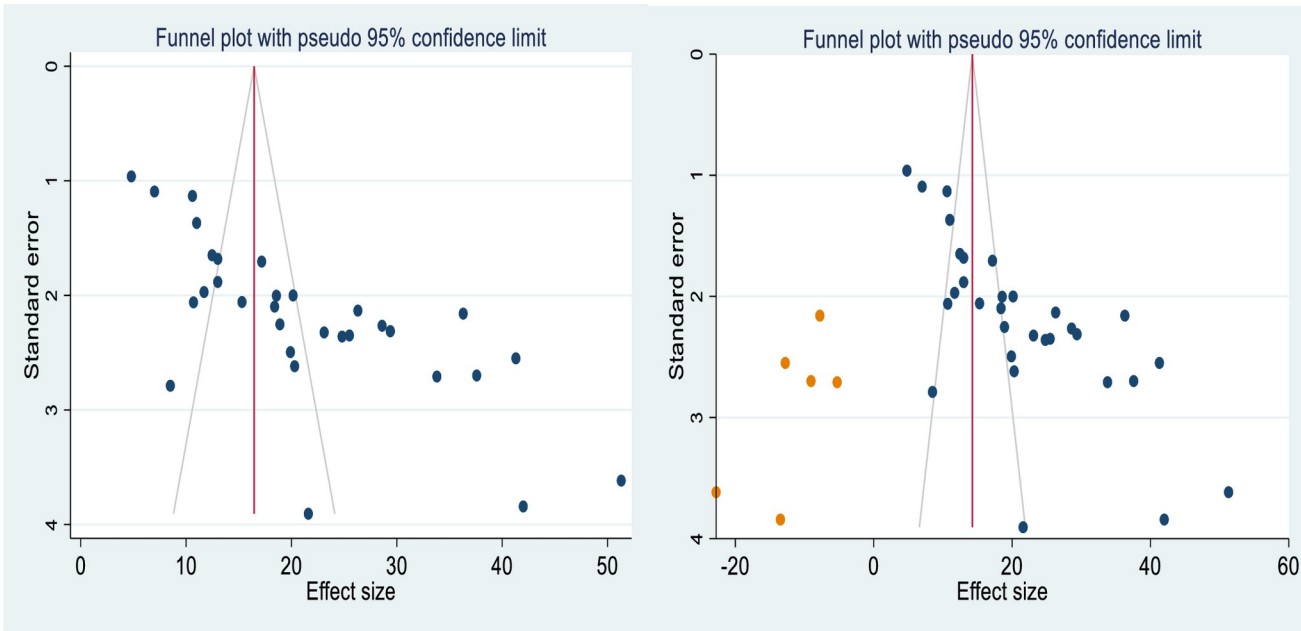

**Fig 3.** Funnel plot before adjustment (3**a**) and after adjustment (3**b**) using trim and fill analysis for publication bias of visual impairment among diabetes patients in Ethiopia, 2023.

subgroup analyses, the heterogeneity of this study might be attributed to differences in study area and period across the included primary studies.

### Factors associated with visual impairment

In the review, fourteen studies [20, 34, 35, 37, 40, 43–47, 49, 50, 59, 60] reported that DM with a duration of diagnosis ≥10 years was significantly associated with visual impairment. The pooled AOR of visual impairment for diabetes patients with a duration of diagnosis ≥10 years was 3.18 (95% CI: 1.85, 5.49; $I^2$ = 91.05%; P<0.001) (Fig 7).

Thirteen studies [35, 37, 45, 46, 49, 50, 55, 58–60, 62–64] showed that the presence of co-morbid hypertension was significantly associated with visual impairment. The pooled AOR of visual impairment for diabetes patients with co-morbid hypertension was 3.26 (95% CI: 1.93, 5.50; $I^2$ = 82.18%; P<0.001) (Fig 8).

Nine studies [20, 34, 35, 39, 46, 49, 50, 63, 64] also reported a significant association between poor glycemic control and visual impairment. The pooled AOR of visual impairment for diabetes patients with poor glycemic control was 4.30 (95% CI: 3.04, 6.06; $I^2$ = 25.51%; P<0.22) (Fig 9).

Thirteen studies [20, 34, 37–40, 43, 45, 47, 55, 56, 59, 63] reported that age ≥56 years was significantly associated with visual impairment. The pooled AOR of visual impairment for diabetes patients with the age of ≥56 years was 4.13 (95% CI: 2.27, 7.52; $I^2$ = 88.82%; P<0.001).

Three studies [37, 59, 63] reported a significant association between family history of DM and visual impairment. The pooled AOR of visual impairment for diabetes patients with family history of DM was 4.18 (95% CI: 2.61, 6.69; $I^2$ = 0%; P<0.98).

Five studies [35, 38, 44, 49, 56] showed that obesity was significantly associated with visual impairment. The pooled AOR of visual impairment for diabetes patients having obesity was 4.77 (95% CI: 3.00, 7.59; $I^2$ = 0%; P<0.93).

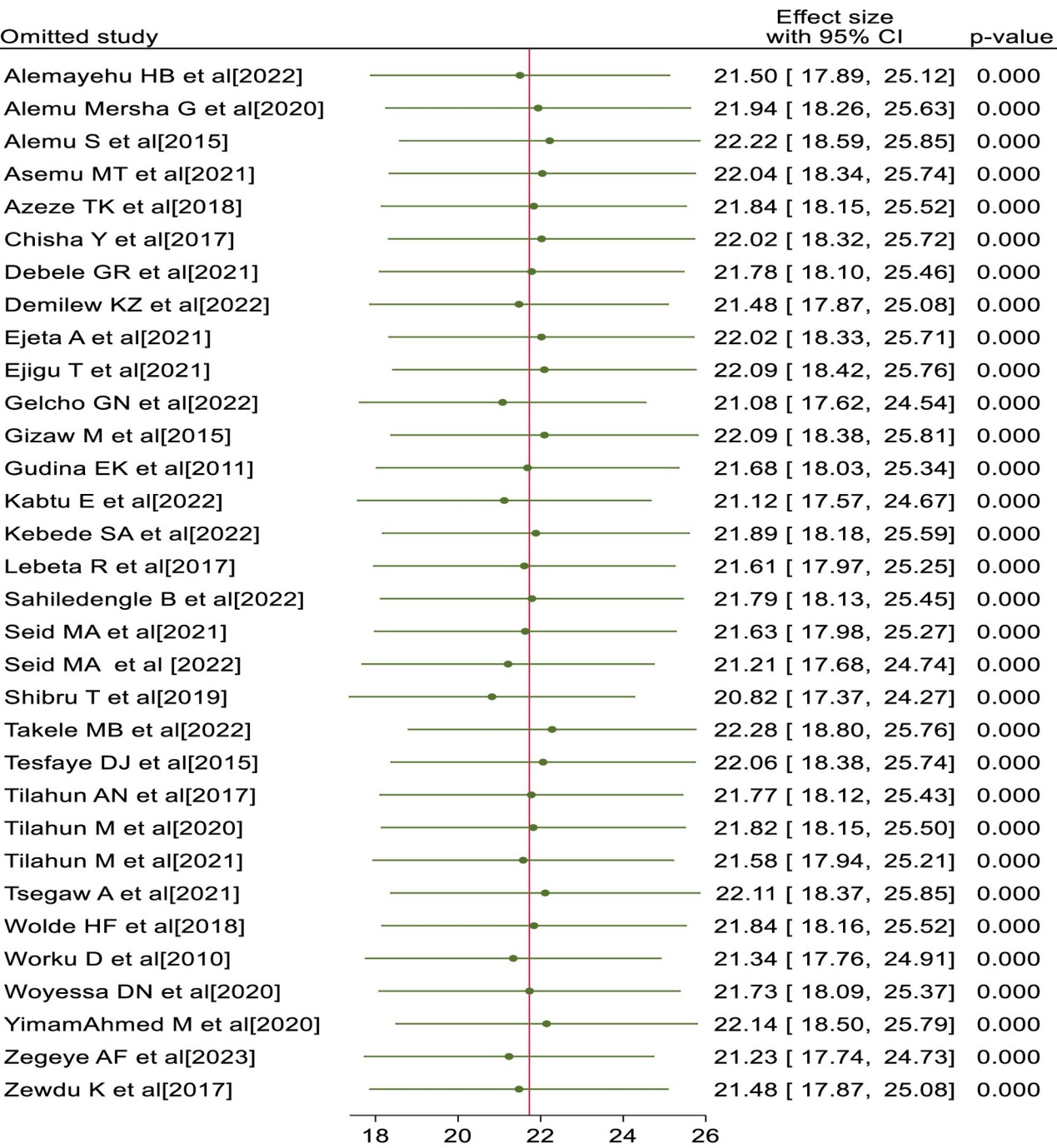

| Omitted study | Effect size with 95% CI | p-value |
|---|---|---|
| Alemayehu HB et al[2022] | 21.50 [ 17.89, 25.12] | 0.000 |
| Alemu Mersha G et al[2020] | 21.94 [ 18.26, 25.63] | 0.000 |
| Alemu S et al[2015] | 22.22 [ 18.59, 25.85] | 0.000 |
| Asemu MT et al[2021] | 22.04 [ 18.34, 25.74] | 0.000 |
| Azeze TK et al[2018] | 21.84 [ 18.15, 25.52] | 0.000 |
| Chisha Y et al[2017] | 22.02 [ 18.32, 25.72] | 0.000 |
| Debele GR et al[2021] | 21.78 [ 18.10, 25.46] | 0.000 |
| Demilew KZ et al[2022] | 21.48 [ 17.87, 25.08] | 0.000 |
| Ejeta A et al[2021] | 22.02 [ 18.33, 25.71] | 0.000 |
| Ejigu T et al[2021] | 22.09 [ 18.42, 25.76] | 0.000 |
| Gelcho GN et al[2022] | 21.08 [ 17.62, 24.54] | 0.000 |
| Gizaw M et al[2015] | 22.09 [ 18.38, 25.81] | 0.000 |
| Gudina EK et al[2011] | 21.68 [ 18.03, 25.34] | 0.000 |
| Kabtu E et al[2022] | 21.12 [ 17.57, 24.67] | 0.000 |
| Kebede SA et al[2022] | 21.89 [ 18.18, 25.59] | 0.000 |
| Lebeta R et al[2017] | 21.61 [ 17.97, 25.25] | 0.000 |
| Sahiledengle B et al[2022] | 21.79 [ 18.13, 25.45] | 0.000 |
| Seid MA et al[2021] | 21.63 [ 17.98, 25.27] | 0.000 |
| Seid MA  et al [2022] | 21.21 [ 17.68, 24.74] | 0.000 |
| Shibru T et al[2019] | 20.82 [ 17.37, 24.27] | 0.000 |
| Takele MB et al[2022] | 22.28 [ 18.80, 25.76] | 0.000 |
| Tesfaye DJ et al[2015] | 22.06 [ 18.38, 25.74] | 0.000 |
| Tilahun AN et al[2017] | 21.77 [ 18.12, 25.43] | 0.000 |
| Tilahun M et al[2020] | 21.82 [ 18.15, 25.50] | 0.000 |
| Tilahun M et al[2021] | 21.58 [ 17.94, 25.21] | 0.000 |
| Tsegaw A et al[2021] | 22.11 [ 18.37, 25.85] | 0.000 |
| Wolde HF et al[2018] | 21.84 [ 18.16, 25.52] | 0.000 |
| Worku D et al[2010] | 21.34 [ 17.76, 24.91] | 0.000 |
| Woyessa DN et al[2020] | 21.73 [ 18.09, 25.37] | 0.000 |
| YimamAhmed M et al[2020] | 22.14 [ 18.50, 25.79] | 0.000 |
| Zegeye AF et al[2023] | 21.23 [ 17.74, 24.73] | 0.000 |
| Zewdu K et al[2017] | 21.48 [ 17.87, 25.08] | 0.000 |

Random-effects DerSimonian–Laird model

**Fig 4. Sensitivity analysis of visual impairment among diabetes patients in Ethiopia, 2023.**

Four studies [20, 38, 56, 64] reported a significant association between poor physical activity and visual impairment. The pooled AOR of visual impairment for diabetes patients with poor physical activity was 2.46 (95% CI: 1.75, 3.46; $I^2$ = 0%; P<0.48).

Two studies [38, 56] reported a significant association between the presence of visual symptoms and visual impairment. The pooled AOR of visual impairment for diabetes patients having visual symptoms was 4.28 (95% CI: 2.73, 6.69; $I^2$ = 0%; P<0.85).

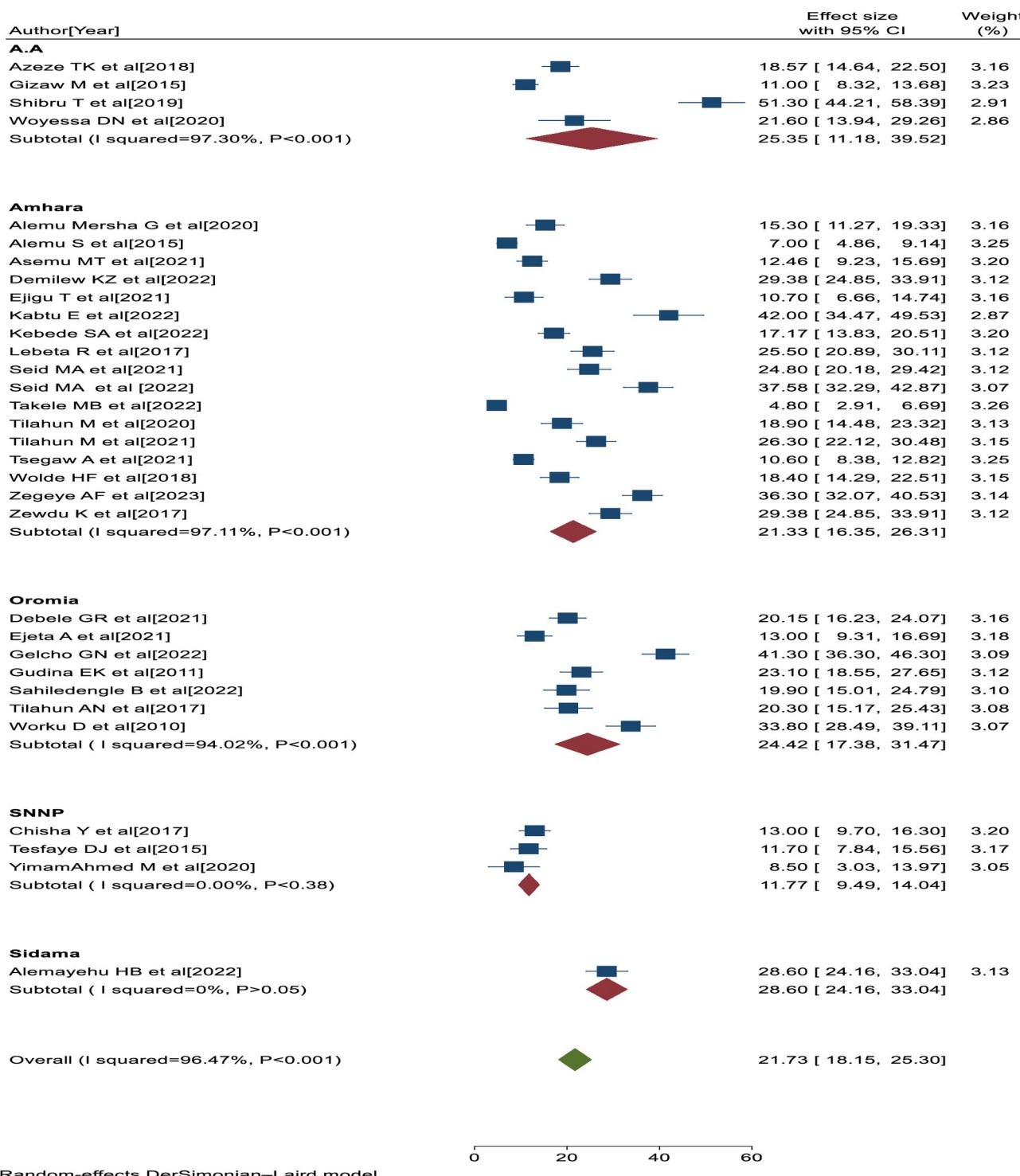

**Fig 5. Forest plot of the prevalence of visual impairment with 95% CIs of the sub-group analysis by study areas.**

Two studies [34, 56] showed that no history of eye exam was significantly associated with visual impairment. The pooled AOR of visual impairment for diabetes patients with no history of eye exam was 2.30 (95% CI: 1.47, 3.57; $I^2$ = 0%; P<0.34).

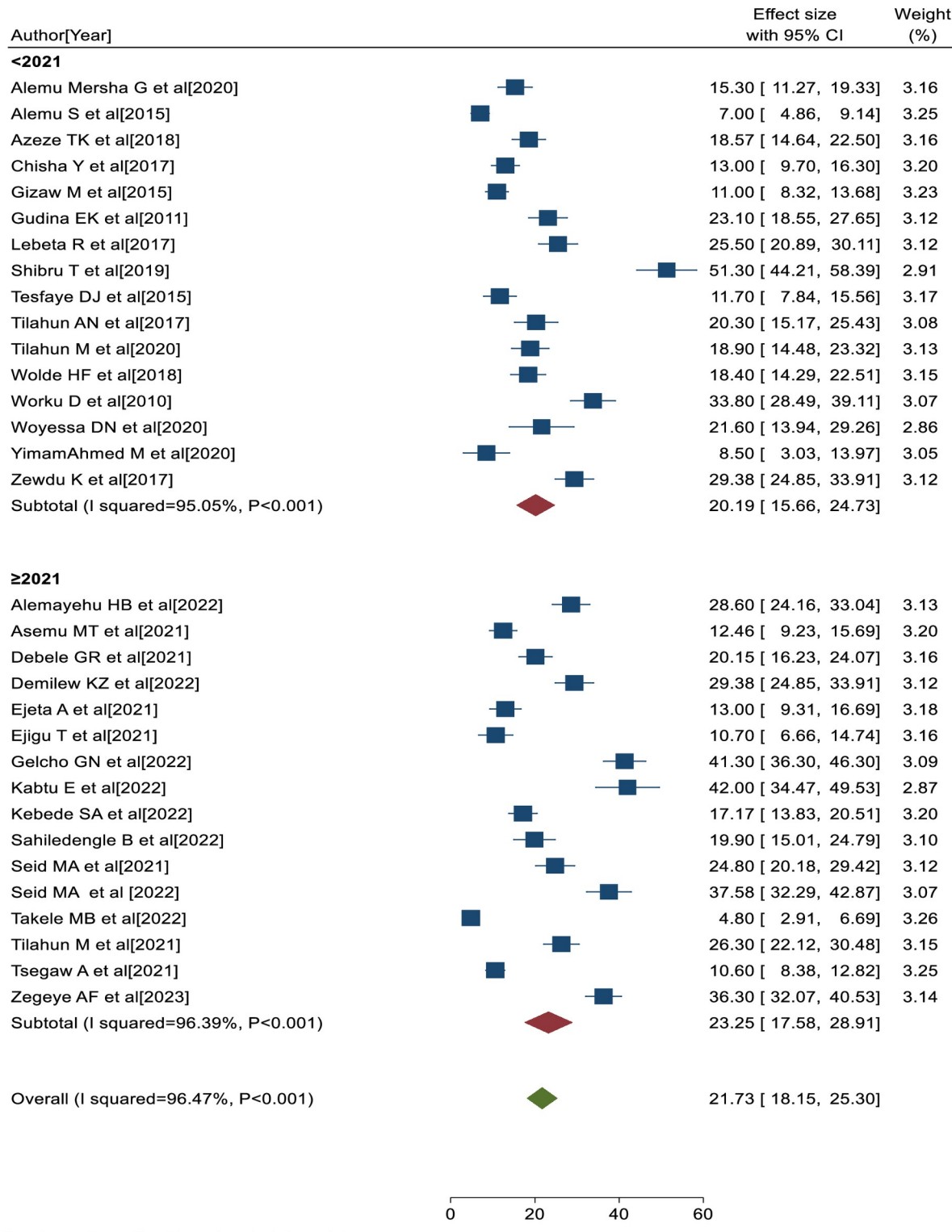

**Fig 6. Forest plot of the prevalence of visual impairment with 95% CIs of the sub-group analysis by study period.**

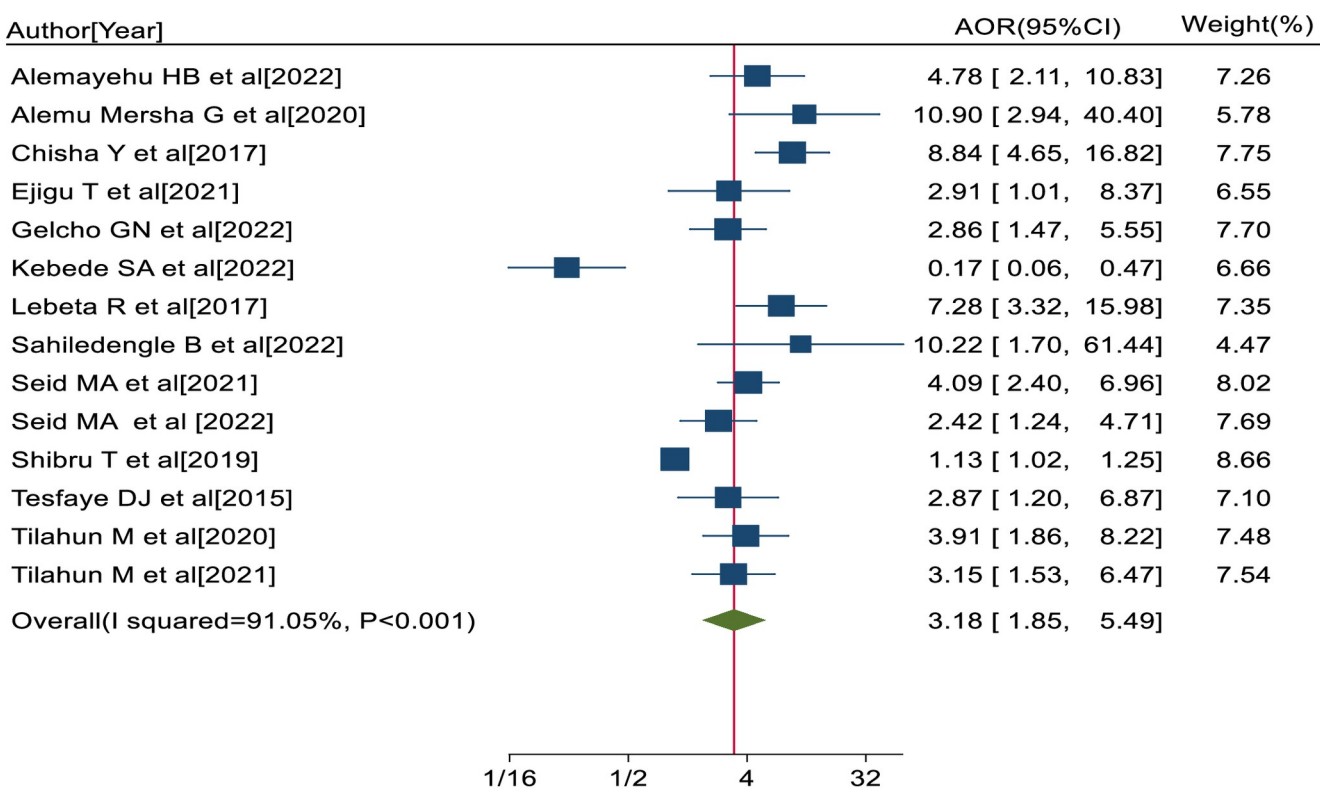

| Author[Year] | AOR(95%CI) | Weight(%) |
|---|---|---|
| Alemayehu HB et al[2022] | 4.78 [ 2.11, 10.83] | 7.26 |
| Alemu Mersha G et al[2020] | 10.90 [ 2.94, 40.40] | 5.78 |
| Chisha Y et al[2017] | 8.84 [ 4.65, 16.82] | 7.75 |
| Ejigu T et al[2021] | 2.91 [ 1.01, 8.37] | 6.55 |
| Gelcho GN et al[2022] | 2.86 [ 1.47, 5.55] | 7.70 |
| Kebede SA et al[2022] | 0.17 [ 0.06, 0.47] | 6.66 |
| Lebeta R et al[2017] | 7.28 [ 3.32, 15.98] | 7.35 |
| Sahiledengle B et al[2022] | 10.22 [ 1.70, 61.44] | 4.47 |
| Seid MA et al[2021] | 4.09 [ 2.40, 6.96] | 8.02 |
| Seid MA et al [2022] | 2.42 [ 1.24, 4.71] | 7.69 |
| Shibru T et al[2019] | 1.13 [ 1.02, 1.25] | 8.66 |
| Tesfaye DJ et al[2015] | 2.87 [ 1.20, 6.87] | 7.10 |
| Tilahun M et al[2020] | 3.91 [ 1.86, 8.22] | 7.48 |
| Tilahun M et al[2021] | 3.15 [ 1.53, 6.47] | 7.54 |
| Overall(I squared=91.05%, P<0.001) | 3.18 [ 1.85, 5.49] | |

Random-effects DerSimonian–Laird model

**Fig 7. Forest plot of the AORs with 95% CIs of studies on the association of DM with duration of diagnosis ≥10 years and visual impairment among diabetes patients in Ethiopia, 2023.**

## Discussion

This review aimed to determine the overall pooled prevalence of visual impairment and its associated factors among diabetes patients in Ethiopia. In this study, the pooled prevalence of visual impairment was 21.73% (95% CI:18.15, 25.30; $I^2$ = 96.47%; P<0.001), which was higher than the study findings conducted in Spain (8.07%) [71], rural India (10.30%) [72], Northwestern Tanzania (10.30%) [73], Malaysia (13.50%) [74], Pakistan (17.60%) [75], Dares Salaam-Tanzania (18.60%) [76] and India (21.70%) [77]. But this finding was lower than the study findings conducted in Tunisia (22.20%) [78], China (23.0%) [79], Tanzania (23.30%) [80], Ghana (24.0%) [81], India (24.90%) [82], Asia (28.0%) [83], Zimbabwe (28.40%) [84], Bangladesh (29.40%) [85], Cameroon (29.70%) [11], Libya (30.60%) [86], China (34.08%) [87], Zambia (36.0%) [12], Nepal (38.26%) [88], Sudan (39.90%) [89], Iran (41.90%) [90] and Yemen (76.50%) [91]. This variation might be due to differences in healthcare systems, methodologies, study settings, study periods, sample sizes and differences in health-seeking behavior of the study participants [26, 55, 57].

Besides, the finding of this study reported that diabetes patients with a duration of diagnosis ≥10 years were 3.18 times more likely to develop visual impairment compared to diabetes patients with a duration of diagnosis <10 years. This finding was congruent with studies conducted in China [13] and India [82]. The likely reason for this association is prolonged diabetes can decrease insulin hormone production by the pancreas or result in target cell resistance. This, in turn, increases the risk of developing diabetic retinopathy, cataract, and ocular edema that cause visual impairment [34, 47, 75, 88].

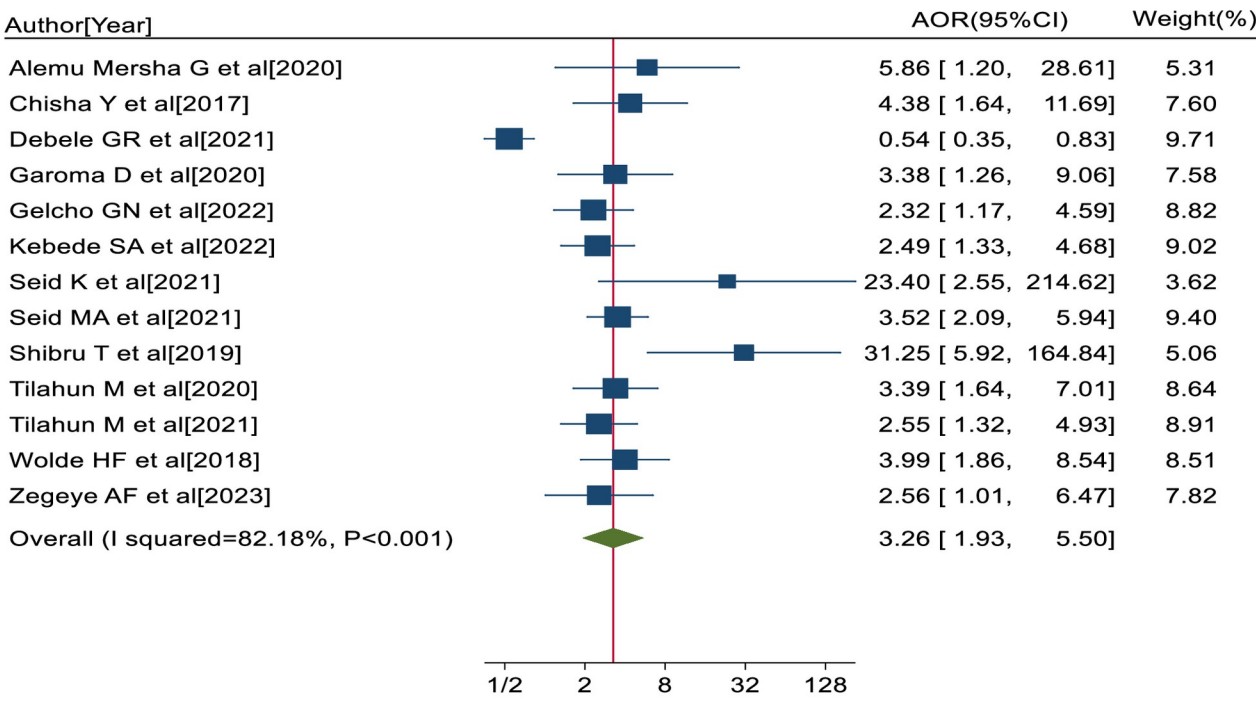

**Fig 8. Forest plot of the adjusted odds ratios with 95% CIs of studies on the association of comorbid hypertension and visual impairment among diabetes patients in Ethiopia, 2023.**

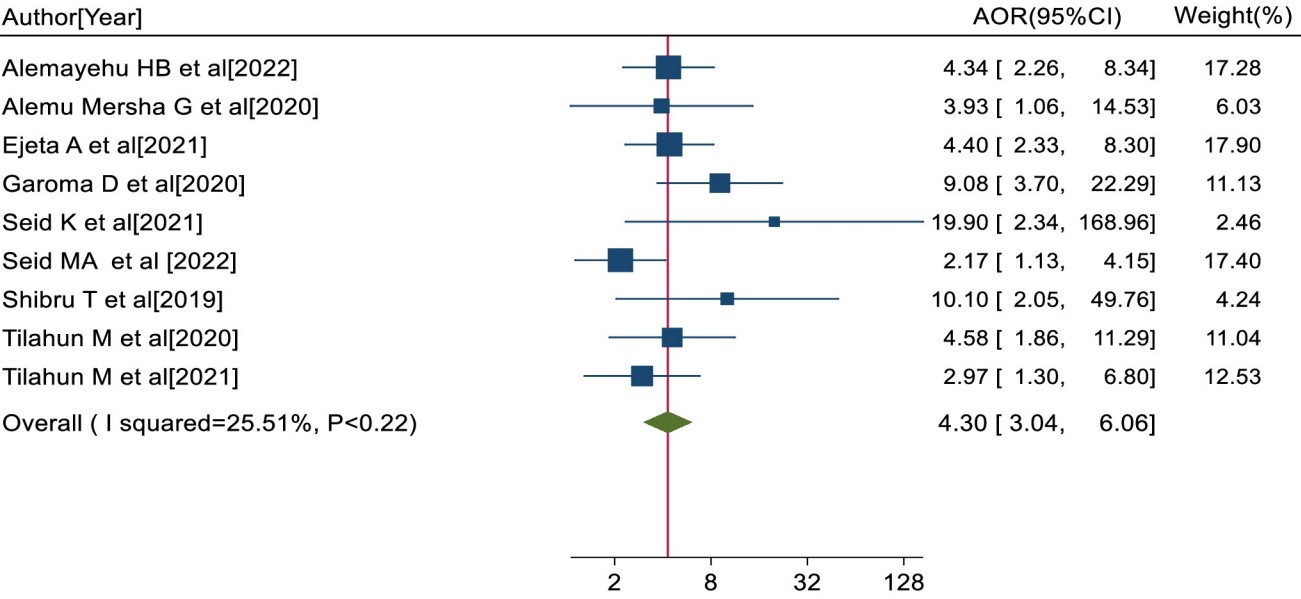

**Fig 9. Forest plot of the adjusted odds ratios with 95% CIs of studies on the association of poor glycemic control and visual impairment among diabetes patients in Ethiopia, 2023.**

The finding of this study also showed that diabetes patients with co-morbid hypertension were 3.26 times more likely to develop visual impairment than diabetes patients without co-morbid hypertension. This finding was similar to a study conducted in India [82]. High blood pressure accelerates the progress and development of micro vascular complications due to increased intracellular hyperglycemia. So, increased plasma glucose level leads to damage to retinal blood vessels and glomeruli, and impairing the regulation of retinal perfusion. Finally, it ends up with visual impairment [20, 82, 88].

Additionally, this study reported that diabetes patients with poor glycemic control were 4.30 times more likely to encounter visual impairment compared to diabetes patients with good glycemic control. This finding was in line with a study conducted in India [82]. It could be explained that an increment in the level of hyperglycemia or having poor glycemic control can increase the onset and rate of progression of diabetic retinopathy, leading to visual impairment [34].

Similarly, the study finding showed that diabetes patients with the age ≥56 years were 4.13 times more likely to develop visual impairment compared to diabetes patients with the age of <56 years. This finding was consistent with a study conducted in Bangladesh [85]. This might be explained as age advances, there might be decrease in physical activity, loss of muscle mass, gain weight and the fatty cells become more resistant to insulin action leading to hyperglycemia. Besides, as age increases, blood vessels become hard, losing their elasticity and more stiffened and leads to cardiac insufficient which end-up with micro vascular complications [20, 39, 63].

Likewise, the study finding indicated that diabetes patients with a family history of DM were 4.18 times more likely to experience visual impairment than diabetes patients who had no family history. This finding was in line with a study conducted in Iran [92]. A family history of diabetes suggests familial genetic and epigenetic contributions to the disease complications. Therefore, patients with a family history of DM are more likely to develop micro vascular complications, such as diabetic retinopathy, cataract and macular edema, leading to visual impairment [59].

In this study, diabetes patients having obesity were also 4.77 times more likely to develop visual impairment compared to patients without having obesity. This finding was similar to a study conducted in Bangladesh [85] and Iran [93]. It could be explained that obesity causes increasing blood viscosity, oxidative stress, vascular growth factors, leptin, cytokines, and intercellular adhesion molecule 1 (ICAM 1), which leads to micro vascular complications and visual impairment [94].

Similarly, the finding of this reported that diabetes patients with poor physical activity were 2.46 times more likely to encounter visual impairment compared to patients with good physical activity. This might be because exercise can promote an increase in the bioavailability of nitric oxide which decreases blood pressure, post-exercise can increase glycolipid uptake and utilization, which improves glucose homeostasis, insulin sensitivity, maintaining glycemic level and optimized body mass index [95].

Additionally, this study indicated that diabetes patients having visual symptoms were 4.28 times more likely to develop visual impairment than patients without visual symptoms. Visual symptoms, such as eye pain, low vision and blurring of vision among diabetes patients can be worsened as the DM advances, leading to visual impairment [38, 56].

Furthermore, the finding of this study reported that diabetes patients having no history of eye exam were 2.30 times more likely to encounter visual impairment compared to their counterparts. This might be due to the fact that the utilization of eye care services for diabetic patients is vital for managing sight-threatening diabetes-related eye complications early. On

the contrary, those diabetes patients who didn't have a history of eye examinations are highly susceptible to undiagnosed diabetes-related eye complications [34, 56].

## Strengths and limitations of the study

To the best of our knowledge, this is the first study to combine the results of multiple studies conducted in Ethiopia, providing stronger evidence on visual impairment and the factors predicting it. While all the studies are of good quality, it should be noted that the majority of the studies analyzed were cross-sectional. Moreover, the study couldn't perform a subgroup analysis using the study designs.

## Conclusions

The overall pooled prevalence of visual impairment was considerably high in Ethiopia. DM with a duration of diagnosis ≥10 years, presence of co-morbid hypertension, poor glycemic control, age ≥56 years, family history of DM, obesity, poor physical activity, presence of visual symptoms and no history of eye exam were independent predictors of visual impairment. Therefore, diabetic patients with these identified risks should be screened, and managed early to reduce the occurrence of visual impairment related to diabetes. Moreover, public health policy with educational programs and regular promotion of sight screening for all diabetes patients is needed.

## Supporting information

**S1 Table. PRISMA statement guideline.**
(DOCX)

**S1 File.**
(DOCX)

## Acknowledgments

We would like to extend our deepest gratitude to Mr. Henok Andualem for his unreserved statistical and methodological support throughout the review.

## Author Contributions

**Conceptualization:** Tigabu Munye Aytenew, Habtamu Shimels Hailemeskel, Yohannes Tesfahun Kassie, Yeshiambaw Eshetie, Shegaw Zeleke, Muluken Chanie Agimas, Amare Simegn.

**Data curation:** Tigabu Munye Aytenew, Solomon Demis Kebede, Amare Kassaw, Netsanet Ejigu, Shegaw Zeleke.

**Formal analysis:** Tigabu Munye Aytenew, Muluken Chanie Agimas.

**Methodology:** Tigabu Munye Aytenew, Binyam Minuye Birhane, Solomon Demis Kebede, Worku Necho Asferie, Habtamu Shimels Hailemeskel, Amare Kassaw, Sintayehu Asnakew, Yohannes Tesfahun Kassie, Gebrehiwot Berie Mekonnen, Melese Kebede, Yeshiambaw Eshetie, Shegaw Zeleke, Muluken Chanie Agimas, Amare Simegn.

**Resources:** Sintayehu Asnakew, Netsanet Ejigu.

**Software:** Tigabu Munye Aytenew, Worku Necho Asferie.

**Validation:** Tigabu Munye Aytenew, Demewoz Kefale, Melese Kebede, Amare Simegn.

**Visualization:** Binyam Minuye Birhane.

**Writing – original draft:** Tigabu Munye Aytenew.

**Writing – review & editing:** Tigabu Munye Aytenew, Demewoz Kefale, Binyam Minuye Birhane, Worku Necho Asferie, Amare Kassaw, Sintayehu Asnakew, Gebrehiwot Berie Mekonnen, Melese Kebede, Shegaw Zeleke, Amare Simegn.

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
