## [Decision Letter · Decision Letter 0]

8 Feb 2024

PONE-D-23-23377Visual impairment and its associated factors among diabetes patients in Ethiopia: Systematic review and Meta-analysisPLOS ONE

Dear Dr. Aytenew,

Thank you for submitting your manuscript to PLOS ONE. After careful consideration, we feel that it has merit but does not fully meet PLOS ONE’s publication criteria as it currently stands. Therefore, we invite you to submit a revised version of the manuscript that addresses the points raised during the review process.

We look forward to receiving your revised manuscript.

Kind regards,

Mohammed Feyisso Shaka, MPH

Academic Editor

PLOS ONE

Journal Requirements:

https://journals.plos.org/plosone/article?id=10.1371%2Fjournal.pone.0276194

https://onlinelibrary.wiley.com/doi/10.1111/tmi.13684

https://dmsjournal.biomedcentral.com/articles/10.1186/s13098-021-00704-w

In your revision ensure you cite all your sources (including your own works), and quote or rephrase any duplicated text outside the methods section. Further consideration is dependent on these concerns being addressed.

Additional Editor Comments:

Please address the peer reviewers comments raised and re submit the revised manuscript following the journal guidelines. Please find the peer reviewers report below. 

Reviewers' comments:

Reviewer's Responses to Questions

**Comments to the Author**

1. Is the manuscript technically sound, and do the data support the conclusions?

Reviewer #1: Partly

Reviewer #2: Yes

2. Has the statistical analysis been performed appropriately and rigorously? 

Reviewer #1: Yes

Reviewer #2: Yes

3. Have the authors made all data underlying the findings in their manuscript fully available?

Reviewer #1: No

Reviewer #2: Yes

4. Is the manuscript presented in an intelligible fashion and written in standard English?

Reviewer #1: No

Reviewer #2: Yes

5. Review Comments to the Author

Reviewer #1: Comments to Author (PONE-D-23-23377)

Thank you, I am happy for the chance to review this kind of manuscript. Here below are my comments on the manuscript.

This is a well-written and important study highlighting Visual impairment and its associated factors among diabetes patients in Ethiopia. There are aspects I have great concern-on topic synthesis

1. In almost all of your individual studies included in your meta-analysis, the topic was “prevalence and associated factors of diabetic retinopathy, but your title said “Visual impairment and its associated factors among diabetes patients in Ethiopia: systematic review and Meta-analysis”. Do you believe that the terms "visual impairment" and "diabetic retinopathy" are similar? I have noticed also an existing systematic review in Ethiopia," Diabetic retinopathy in Ethiopia: A systematic review and meta-analysis. It seems your results are overlapping, including, in part, same studies.”https://www.sciencedirect.com/science/article/abs/pii/S1871402119302036?via%3Dihub.

In addition, when we see your discussion, the literatures used was focuses on diabetic retinopathy rather than on visual impairment and associated factors. Are you going to compare unrelated studies?

2. Before resubmission, a person proficient in written English edits the manuscript. It is important that the message being conveyed in the manuscript is as unambiguous as possible. For instance, here are some examples of language related problems.

Abstract

Page-2 -rewrite the vague sentence “Although numerous attempts have been made to ascertain the occurrence of visual impairment and its associated factors among diabetes patients in Ethiopia, several original studies have reported incongruous results” to

Page-3-Change literatures to literature

Page-3 change ‘from’ to on

Page 3-4 change is to are (patients is needed).

P 3 Line 66-67 please rewrite or remove these sentence in your abstract conclusion i.e. “Therefore, diabetic patients should engage in regular physical activity, control their glycemic level with proper use of medications, manage co morbid hypertension, and have regular eye exams”.

3. Page 4 Line 85 to 86, I believe there are more recent sources, for instance, https://www.researchgate.net/publication/372489279_The_burden_of_visual_impairment_among_Ethiopian_adult_population_Systematic_review_and_meta-analysis

4. Why you limited the numbers of keywords for your search? Especially related to individuals studies included in your references like “retinopathy, diabetic retinopathy and cataract etc.”

Reviewer #2: The authors have produced an excellent paper by effectively implementing a sound procedure and executing appropriate steps to address the encountered problem. Here are my concerns

1.The study by Abere and colleagues, outlined in their paper on the determinants of diabetic retinopathy, uncovered a factor linked to this condition. What unique contribution does your research bring to the table in terms of novelty?

2.Multiple cross-sectional prevalence studies have been conducted on this topic within the country. What specific contribution do you aim to make to the scientific community with your research?

3.What is the criteria to say that the level of visual impairment is high?

4. Poor definition of visual impairment you have to had a scientific definition ?

5. Where is supplemental table 3 ( risk of bias assessment ) ?

6. Improve English language editing

6. PLOS authors have the option to publish the peer review history of their article (what does this mean?). If published, this will include your full peer review and any attached files.

Reviewer #1: No

Reviewer #2: No

---

## [Author Response · Author response to Decision Letter 0]

12 Feb 2024

Dear Editors and reviewers:

We sincerely appreciate the valuable comments and suggestions you raised. The thorough review helped immensely in the shaping of the manuscript. The comments and suggestions have been closely followed and revisions have been made accordingly. The following are the questions that have been extracted from the Editor and Reviewers’ comments along with our summarized responses. Thank you very much for your constructive comments. We tried to inculcate your comments and questions as described below. The changes will be attached with

Title: Visual impairment among diabetes patients in Ethiopia: A systematic review and meta-analysis.

Authors:

TMA: tigabumunye21@gmail.com

DK: demewozk@yahoo.com

BM: biniamminuye@yahoo.com

SD: solomondemis@gmail.com

WN: workunecho@gmail.com

HS: habtamushimels21@gmail.com

AK: amarekassaw2009@gmail.com

SA: sintie579@gmail.com

YT: tesfahunyohannes08@gmail.com

GB: geberehwot2004@gmail.com

MK: meleske143@gmail.com

YE: yeshiambaweshetie@gmail.com

NE: netsanet01616@gmail.com

SZ: shegawzn@gmail.com

MC: mulukensrc12@gmail.com

AS: amaresimegn99@gmail.com

Editor Comment #01: Please ensure that your manuscript meets PLOS ONE's style requirements, including those for file naming. The PLOS ONE style templates can be found at https://journals.plos.org/plosone/s/file?id=wjVg/PLOSOne_formatting_sample_main_body.pdf and https://journals.plos.org/plosone/s/file?id=ba62/PLOSOne_formatting_sample_title_authors_affiliations.pdf

Authors’ response: Recognizing your comment, we have looked at the PLOS ONE style templates using the given link and we have ensured that our manuscript meets PLOS ONE's style requirements, including those for file naming. The requested corrections have been included throughout the revised version of the manuscript.

Editor Comment #02: Note from Emily Chenette, Editor in Chief of PLOS ONE, and Iain Hrynaszkiewicz, Director of Open Research Solutions at PLOS: Did you know that depositing data in a repository is associated with up to a 25% citation advantage (https://doi.org/10.1371/journal.pone.0230416)? If you’ve not already done so, consider depositing your raw data in a repository to ensure your work is read, appreciated and cited by the largest possible audience. You’ll also earn an Accessible Data icon on your published paper if you deposit your data in any participating repository (https://plos.org/open-science/open-data/#accessible-data). 

Authors’ response: We are convinced of this comment, but we faced technical problem while trying to depose the data in to the repository. 

Editor Comment #03: We noticed you have some minor occurrence of overlapping text with the following previous publication(s), which needs to be addressed:

https://journals.plos.org/plosone/article?id=10.1371%2Fjournal.pone.0276194

https://onlinelibrary.wiley.com/doi/10.1111/tmi.13684

https://dmsjournal.biomedcentral.com/articles/10.1186/s13098-021-00704-w

In your revision ensure you cite all your sources (including your own works), and quote or rephrase any duplicated text outside the methods section.

Authors’ response: Accepting your constructive comment, we have paraphrased the overlapping texts throughout the manuscript. 

Editor Comment #04: Please include captions for your supporting Information files at the end of your manuscript, and update any in-text citations to match accordingly. Please see our Supporting Information guidelines for more information: http://journals.plos.org/plosone/s/supporting-information. 

Authors’ response: Thank you! We have put the supporting table captions at the end of the manuscript.

Editor Comment #05: While revising your submission; please upload your figure files to the Preflight Analysis and Conversion Engine (PACE) digital diagnostic tool, https://pacev2.apexcovantage.com/. PACE helps ensure that figures meet PLOS requirements. To use PACE, you must first register as a user. Registration is free. Then, login and navigate to the UPLOAD tab, where you will find detailed instructions on how to use the tool. If you encounter any issues or have any questions when using PACE, please email PLOS at figures@plos.org. Please note that Supporting Information files do not need this step.

Authors’ response: Recognizing your feedback, we have uploaded our figures using the Preflight Analysis and Conversion Engine (PACE) digital diagnostic tool. Reviewer # 1:

Reviewer #1 comment and suggestion #01: In almost all of your individual studies included in your meta-analysis, the topic was “prevalence and associated factors of diabetic retinopathy, but your title said “Visual impairment and its associated factors among diabetes patients in Ethiopia: systematic review and Meta-analysis”. Do you believe that the terms "visual impairment" and "diabetic retinopathy" are similar? 

I have noticed also an existing systematic review in Ethiopia," Diabetic retinopathy in Ethiopia: A systematic review and meta-analysis. It seems your results are overlapping, including, in part, same studies.”https://www.sciencedirect.com/science/article/abs/pii/S1871402119302036?via%3Dihub. In addition, when we see your discussion, the literatures used were focuses on diabetic retinopathy rather than on visual impairment and associated factors. Are you going to compare unrelated studies?

Authors’ response: Thank you for your critical view of our manuscript! The terms "visual impairment" and "diabetic retinopathy" are not exactly similar, that means visual impairment is inclusive. In addition to DM, visual impairment might also occur following congenitally, increased age, trauma, etc. However, diabetic retinopathy occurs only when the DM become complicated (chronic complication). But, both of them are characterized with functional impairment of vision. Therefore, they are related. On the other hand, our results were not overlapped with the stated study entitled with "Diabetic retinopathy in Ethiopia: A systematic review and meta-analysis" Because, this study focused only on the pooled prevalence of diabetic retinopathy excluding assessing the possible pooled risk factors , whereas our study focused on both the pooled prevalence and risk factors. Additionally, the first study was conducted before five years ago by including around eleven eligible primary studies, but the current study was conducted by including thirty-four eligible primary studies. i.e. Numerous studies were not included in the first study were included in the current one. 

Reviewer #1 comment and suggestion #02: Before resubmission, a person proficient in written English edits the manuscript. It is important that the message being conveyed in the manuscript is as unambiguous as possible. For instance, here are some examples of language related problems.

Abstract

Page-2 -rewrite the vague sentence “Although numerous attempts have been made to ascertain the occurrence of visual impairment and its associated factors among diabetes patients in Ethiopia, several original studies have reported incongruous results” to

Page-3: Change ‘literatures’ to literature

Page-3: Change ‘from’ to on

Page 3-4: Change ‘is’ to are (patients is needed).

Page 3 Line 66-67 please rewrite or remove these sentence in your abstract conclusion i.e. “Therefore, diabetic patients should engage in regular physical activity, control their glycemic level with proper use of medications, manage co morbid hypertension, and have regular eye exams”.

Authors’ response: Thank you for your critical view! We have edited these typological errors accordingly in the manuscript.

Reviewer #1 comment and suggestion #03: 

Page 4 Line 85 to 86, I believe there are more recent sources, for instance, https://www.researchgate.net/publication/372489279_The_burden_of_visual_impairment_among_Ethiopian_adult_population_Systematic_review_and_meta-analysis

Authors’ response: Thank you for your feedback! But, the stated study was focused on the general population (all adult population), whereas our study was focused on only adult population with DM i.e. different target population.

Reviewer #1 comment and suggestion #04: Why you limited the numbers of keywords for your search? Especially related to individuals studies included in your references like “retinopathy, diabetic retinopathy and cataract etc”

Authors’ response: Thank you very much for your constructive comment! We have revised the search section of the manuscript based on the given direction.

Reviewer # 2:

Reviewer # 2 comment and suggestion #01: .The study by Abere and colleagues, outlined in their paper on the determinants of diabetic retinopathy, uncovered a factor linked to this condition. What unique contribution does your research bring to the table in terms of novelty?

Authors’ response: Thank you for your critical view! The study conducted by Abere and his colleagues tried to appreciate only the pooled determinants, whereas our study tried to appreciate both the pooled prevalence and determinants.

Reviewer #2 comment and suggestion #02: Multiple cross-sectional prevalence studies have been conducted on this topic within the country. What specific contribution do you aim to make to the scientific community with your research?

Authors’ response: Sure! Even though several cross-sectional prevalence studies were conducted on this topic within the country, their findings were inconsistent. Therefore, this study was used to aggregate these studies, giving stronger evidence on this topic in the country.

Reviewer #2 comment and suggestion #03: What is the criterion to say that the level of visual impairment is high? 

Authors’ response: Thank you! The parameter was by considering/comparing the 95%CI of the current and previous study findings. 

Reviewer #2 comment and suggestion #04: Poor definition of visual impairment you have to have a scientific definition?

Authors’ response: Accepting of your valuable comment, we have revised the operational definition of visual impairment.

Reviewer #2 comment and suggestion #05: Where is supplemental table 3 (risk of bias assessment)?

Authors’ response: Thank you for your valuable comment, it was under supplemental file 2, and we have put it separately as “Supplemental Table 3” now in the manuscript.

Reviewer #2 comment and suggestion #06: Improve English language editing.

Authors’ response: Thank you for your constructive comment! We have intensively edited all the editorial problems throughout the manuscript.

---

## [Decision Letter · Decision Letter 1]

24 Apr 2024

Visual impairment among diabetes patients in Ethiopia: A systematic review and meta-analysis

PONE-D-23-23377R1

Dear Dr. Aytenew,

We’re pleased to inform you that your manuscript has been judged scientifically suitable for publication and will be formally accepted for publication once it meets all outstanding technical requirements.

Kind regards,

Mohammed Feyisso Shaka, MPH

Academic Editor

PLOS ONE

Additional Editor Comments (optional):

Reviewers' comments:

Reviewer's Responses to Questions

**Comments to the Author**

1. If the authors have adequately addressed your comments raised in a previous round of review and you feel that this manuscript is now acceptable for publication, you may indicate that here to bypass the “Comments to the Author” section, enter your conflict of interest statement in the “Confidential to Editor” section, and submit your "Accept" recommendation.

Reviewer #1: All comments have been addressed

2. Is the manuscript technically sound, and do the data support the conclusions?

Reviewer #1: Partly

3. Has the statistical analysis been performed appropriately and rigorously? 

Reviewer #1: Yes

4. Have the authors made all data underlying the findings in their manuscript fully available?

Reviewer #1: Yes

5. Is the manuscript presented in an intelligible fashion and written in standard English?

Reviewer #1: Yes

6. Review Comments to the Author

Reviewer #1: (No Response)

7. PLOS authors have the option to publish the peer review history of their article (what does this mean?). If published, this will include your full peer review and any attached files.

Reviewer #1: **Yes: **Dilnessa Fentie

---

## [Editor Report · Acceptance letter]

19 May 2024

PONE-D-23-23377R1 

PLOS ONE

Dear Dr. Aytenew, 

I'm pleased to inform you that your manuscript has been deemed suitable for publication in PLOS ONE. Congratulations! Your manuscript is now being handed over to our production team.

Kind regards, 

on behalf of

Mr. Mohammed Feyisso Shaka 

Academic Editor

PLOS ONE